# TinyFallNet: A Lightweight Pre-Impact Fall Detection Model

**DOI:** 10.3390/s23208459

**Published:** 2023-10-14

**Authors:** Bummo Koo, Xiaoqun Yu, Seunghee Lee, Sumin Yang, Dongkwon Kim, Shuping Xiong, Youngho Kim

**Affiliations:** 1Department of Biomedical Engineering, Yonsei University, Wonju 26493, Republic of Korea; koobummo726@gamil.com (B.K.); fhrm502@yonsei.ac.kr (S.L.); abbey0909@yonsei.ac.kr (S.Y.); aas0851@gmail.com (D.K.); 2Department of Industrial Design, School of Mechanical Engineering, Southeast University, Nanjing 211189, China; xiaoqunyu@seu.edu.cn; 3Department of Industrial and Systems Engineering, Korea Advanced Institute of Science and Technology (KAIST), Daejeon 34141, Republic of Korea; shupingx@kaist.ac.kr

**Keywords:** pre-impact fall detection, lightweight, ConvLSTM, TinyFallNet

## Abstract

Falls represent a significant health concern for the elderly. While studies on deep learning-based preimpact fall detection have been conducted to mitigate fall-related injuries, additional efforts are needed for embedding in microcomputer units (MCUs). In this study, ConvLSTM, the state-of-the-art model, was benchmarked, and we attempted to lightweight it by leveraging features from image-classification models VGGNet and ResNet while maintaining performance for wearable airbags. The models were developed and evaluated using data from young subjects in the KFall public dataset based on an inertial measurement unit (IMU), leading to the proposal of TinyFallNet based on ResNet. Despite exhibiting higher accuracy (97.37% < 98.00%) than the benchmarked ConvLSTM, the proposed model requires lower memory (1.58 MB > 0.70 MB). Additionally, data on the elderly from the fall data of the FARSEEING dataset and activities of daily living (ADLs) data of the KFall dataset were analyzed for algorithm validation. This study demonstrated the applicability of image-classification models to preimpact fall detection using IMU and showed that additional tuning for lightweighting is possible due to the different data types. This research is expected to contribute to the lightweighting of deep learning models based on IMU and the development of applications based on IMU data.

## 1. Introduction

Falls are a serious threat to the safety and well-being of older adults, with the potential to cause severe injuries and even death [1]. Older adults are more vulnerable to falls due to the age-related reduction in muscle mass, decreased bone density, and impaired vision or hearing [2]. The annual incidence of falls in individuals aged over 65 was reported to be over 25%, with the rate increasing to 32–42% for those aged over 70 years old [3]. Statistics on elderly fall cases indicate that falls can result in major injuries (40–60%), minor injuries (30–40%), and fractures (5–6%). The fractures can be fatal: for instance, 20% of fall-related hip fractures result in death within a year [4].

Some studies have tried to detect falls using deep learning. Syed et al. [5] proposed a Convolutional Neural Network (CNN)-XGB Network for fall detection and activity recognition, considering the direction and severity of falls, achieving an unweighted average recall of 88%. Iguchi et al. [6] employed a Convolutional Autoencoder for unsupervised fall detection, demonstrating an accuracy of 98%. Zhang et al. [7] used CNN to recognize 17 movements, including falls, achieving an identification rate of 91.5%. These studies focus on post-fall detection, including impact values during falls. The binary classification accuracy is nearly 100%, with ongoing research in the direction of multi-class classification. However, post-fall detection algorithms can quickly detect falls through alarms, but they cannot prevent the impacts of falls themselves.

The wearable airbag is a solution to prevent fractures caused by falls [8]. It would detect falls during descent through preimpact fall detection and mitigate the impact. The preimpact fall-detection algorithm is a more challenging field compared to post-fall detection, as it involves detecting falls before impact solely based on preimpact signals. Several researchers have developed preimpact fall detection algorithms based on threshold-based methods [8,9,10]. They measured three-axis accelerations and three-axis angular velocities using IMU and calculated significant features (vertical velocity, vertical angle, etc.) to detect falls. Falls are detected if certain fall-detection indicators exceed a pre-defined threshold. They showed good performances based on their specific databases. However, when Ahn et al. [11] applied their algorithm to the SisFall public dataset [12], the specificity decreased by approximately 20–30%.

The development of preimpact fall detection using deep learning has been made possible by the rapid advancement of deep learning techniques, powerful hardware, and publicly available big data. Existing public fall databases, such as SisFall [12], MobiFall [13], and FallAllD [14], are only suitable for post-fall detection rather than preimpact fall detection. They only provide IMU data and do not include video files due to the privacy issue. The specific labels, which include temporal information such as the onset and offset of a fall, were required to train the deep learning model. Musci et al. [15] established their own criteria for temporal annotation by comparing representative video clips from the SisFall dataset with corresponding sensor data recordings. They labeled all the data into three classes (FALL, ALERT, and BKG) and made the labeled data publicly available. In addition, they proposed an LSTM-based model for preimpact fall detection. The accuracies for BKG, ALERT, and FALL were reported as 94.14%, 92.86%, and 97.16%, respectively. Torti et al. [16] optimized LSTM model to embed in the MCU using the SisFall dataset. They achieved accuracies of 93.12%, 90.93%, and 98.33% for BKG, ALERT, and FALL, respectively. Yu et al. [17] proposed the ConvLSTM model to detect preimpact falls. When they trained and tested the model on the SisFall dataset, they showed accuracies of 93.22%, 94.48%, and 98.66% for BKG, ALERT, and FALL, respectively. However, since the labeling in the SisFall dataset may not be accurate, these methods cannot provide a fundamental solution and researchers were not able to provide a lead time. Yu et al. [18] developed a new dataset called “KFall” for preimpact fall detection. This new dataset consisted of three-axis accelerations, three-axis angular velocities, and three-axis Euler angles, which were collected from 32 Korean participants performing 21 ADLs and 15 falls. They provided the moment of fall onset by integrating information from sensor and video data. They trained and evaluated a ConvLSTM model on the KFall dataset, reporting a sensitivity of 99.32%, a specificity of 99.01%, and a lead time of 403 ± 163 ms. ConvLSTM model is currently considered the state-of-the-art model in preimpact fall detection. However, the model ultimately uses nine-axis data (three-axis accelerations, three-axis angular velocities, and three-axis Euler angles), which may not be practical for commercial and wearable airbag applications, where six-axis (three-axis accelerations and three-axis angular velocities) signals are preferred. Moreover, an additional optimization is required to make the model lightweight for embedding in an MCU. Therefore, the ConvLSTM model was benchmarked with the goal of such optimization.

In order to streamline the ConvLSTM model, the architecture needs to be simplified while maintaining its performance. CNN-based classifiers have been primarily used for image processing, so attention was given to models used in image classification. VGGNet [19] could enhance performance while maintaining the output size by increasing the number of layers instead of reducing the filter size. On the other hand, ResNet [20] could decrease the computational load and increase the number of layers through its bottleneck structure, thus delivering high performance with less memory compared to VGGNet. Although it was uncertain whether these models, developed for image classification, would exhibit similar effectiveness for fall detection, Alarifi and Alwadain [21] demonstrated a good performance in post-fall detection using the deep learning model AlexNet, which was previously applied to image classification.

Generally, data collected from young adults are used in algorithm development, due to the potential risks associated with simulated experiments. However, field tests are necessary, as the actual target group is older adults. The extended KFall dataset [22] provides ADL data from older adults, and the Farseeing dataset [23] provides fall data from older adults through long-term monitoring.

In this work, we present the lightweighting of a preimpact fall-detection algorithm using deep learning models commonly employed for image classification for wearable airbags. ConvLSTM, a state-of-the-art model for preimpact fall detection, was benchmarked, and attempts were made to lighten it while maintaining performance. In the pursuit of lightweighting the model, VGGNet and ResNet, which are commonly used image classification models, were applied and analyzed for the preimpact fall-detection problem. As a result, TinyFallNet, based on ResNet, was proposed. The principal contributions of the present work are as follows: Firstly, we demonstrated the applicability of models developed for image classification to the IMU-based preimpact fall-detection problem. Secondly, we demonstrated additional tuning possibilities for image classification models. VGGNet and ResNet require fewer parameters in our problem than the image problem due to the difference in the data type. In particular, ResNet showed that performance was maintained despite removing the Identity Block. Thirdly, the ConvLSTM model was successfully lightweighted while maintaining performance. The lightweight model was named TinyFallNet, and it showed higher accuracy (98.00%) and required smaller memory (0.70 MB) compared to the benchmark ConvLSTM model (accuracy: 97.37%; memory: 1.58 MB). Finally, the proposed TinyFallNet applied well to real-fall situations. The Farseeing dataset was employed to evaluate actual fall movements in the elderly, and the extended KFall dataset was used to assess the ADLs of the elderly. The results yielded a sensitivity of 86.67%, a specificity of 97.97%, and a lead time of 477.7 ± 5.8 ms.

## 2. Materials and Methods

### 2.1. Overall Flowchart

Figure 1 illustrates the overall flowchart of this study. Firstly, the dataset for model development and evaluation was prepared. In the next step, a model was implemented using only the data from young participants in the KFall dataset [18], and efforts were made to lightweight the model. TinyFallNet was proposed accordingly. Finally, the model was evaluated using the elderly ADL data from the KFall dataset [22] and 15 actual fall datasets from the FARSEEING dataset [23].

### 2.2. Public Dataset

The KFall dataset [18,22] was used in this study for two reasons: firstly, it was the first public dataset suitable for preimpact fall detection due to the fact that it provides the time information about the onset and offset of the fall; secondly, ConvLSTM, which served as our benchmark model, was developed using KFall. The dataset was collected from 32 healthy young males (age: 24.9 ± 3.7 years; height: 174.0 ± 6.3 cm; weight: 69.3 ± 9.5 kg) and 10 (five males, five females) community-dwelling old subjects (age: 80.8 ± 2.5 years; height: 166.6 ± 11.9 cm; weight: 65.1 ± 7.2 kg) using an IMU sensor attached to the low back of each subject and sampled at 100 Hz. It consisted of 9-axis IMU data, including 3-axis accelerations, 3-axis angular velocities, and 3-axis Euler angles, along with temporal labels that indicated the fall-onset moment and the fall-impact moment based on synchronized videos. The dataset also included 21 activities of daily living (ADLs) and 15 simulated falls (Table 1 and Table 2). Each activity was performed for a repeated number of trials. Young subjects were required to perform both ADLs and falls, while old subjects did not perform any falls or several ADLs involving risk, including D04, D10, D15 and D21. Additionally, some old participants skipped some ADLs due to limited physical ability.

The FARSEEING dataset [23], currently the only existing dataset containing real-world falls of the elderly, consists of 22 files. Each file incorporates one fall event within 1200 s of data. Out of these, only 15 files were used in this study due to their similar sensor location and sampling frequency to the KFall dataset (Table 3). The falls were collected from eight elderly subjects (two males and six females) with an average age of 66.9 ± 6.5 years, an average height of 162.2 ± 9.3 cm, and average weight of 74.2 ± 10.3 kg.

### 2.3. Data Preparation

The data-preparation process followed earlier studies on the benchmark model ConvLSTM [17,18,22]. The sliding-window technique with a window size of 0.5 s and a step size of 0.1 s was applied to the motion file along the timestamps for the development of deep learning models [24,25]. Each window was labeled as the non-fall or fall class to support supervised learning. For ADL tasks, all windows were labeled as the non-fall class. For windows with at least 40% of readings within the fall phase (from the moment of fall onset to the moment of fall impact), they were labeled as fall class, while the remaining windows were also labeled as the non-fall class.

### 2.4. Deep Learning Models

The three deep learning models (ConvLSTM, VGGNet, and ResNet) were applied in the study. 

#### 2.4.1. ConvLSTM

ConvLSTM model is the state-of-the-art model in preimpact fall detection [17,18,22]. The model combines the strengths of Convolutional Neural Networks (CNNs) and long short-term memory (LSTM) networks, enabling it to capture both local dependencies and long-term temporal relationships in human motion data. The architecture of the ConvLSTM model consists of three convolutional layers and two LSTM layers, each with dropout operations. The convolutional layers consist of Conv1D layer (number of filters: 64; kernel size: 7; strides: 1), batch normalization, ReLU activation, and max pooling operations (pool size: 2). The LSTM layers have an output channel of 64 and a dropout rate of 0.5.

#### 2.4.2. VGGNet

VGGNet [19] is a deep CNN model developed by researchers at the University of Oxford’s Vision Geometry Group and Google’s DeepMind. It achieves state-of-the-art performance by stacking multiple convolutional layers, typically 16 to 19 layers deep, with 3 × 3 convolutional kernels and 2 × 2 maximum pooling layers. This design enables VGGNet to achieve a significant drop-in error rate, leading to second place in the ILSVRC 2014 competition. The original model used Conv2D layers, but Conv1D layers were applied in the study considering that our data type was time-series data rather than image data.

#### 2.4.3. ResNet

ResNet [20] was developed to address the issue of vanishing gradients in deep neural networks, which can occur when training a network with a large number of layers. The ResNet architecture uses residual blocks, which enable the training of very deep neural networks. In a residual block, the output of a layer is added to its input through a shortcut connection, allowing gradients to flow directly through the block. This helps prevent the vanishing gradient problem and enables the training of deeper networks. ResNet has been widely used in computer vision tasks, such as image classification and object detection, and has achieved state-of-the-art performance on several benchmark datasets. The ResNet architecture comes in different variants, including ResNet-18, ResNet-34, ResNet-50, ResNet-101, and ResNet-152, with each variant having a different number of layers and complexity of the network. The original model used Conv2D layers, but Conv1D layers were applied in the study considering that our data type was time-series data rather than image data.

### 2.5. Model Training

The deep learning models were implemented using the TensorFlow 2.0 library on a Windows 10 (64-bit) computer equipped with a 3.4 GHz CPU i7-13700K, 64 GB RAM, and an Nvidia GeForce RTX 3080 GPU card. The models were trained and tested on this computer using six-dimensional data (three-axis accelerations and three-axis angular velocities) with a window size of 50 frames (0.5 s) as input. A sufficient number of data patterns is essential to train the deep learning model. Therefore, only data from young participants (32 subjects) in the KFall dataset were used for model development. Data from elderly individuals (10 subjects) in KFall were reserved solely for the evaluation of the finally proposed model. Following general guidelines, 80% of the data (26 subjects) were used for training, while the remaining 20% (6 subjects) were reserved for testing. The batch size was set to 64, and the total number of epochs was 200, as described by previous research [22]. To prevent overfitting, early stopping was applied during training, with a validation ratio of 20%. Training was stopped if the validation accuracy was not improved for 20 consecutive epochs. 

### 2.6. Performance Evaluation

The performance of deep learning models was evaluated in terms of sensitivity, specificity, accuracy, lead time, and memory size. For performance evaluation, the values of sensitivity, specificity, and accuracy were calculated as follows: (1)Sensitivity=TP/(TP+FN)×100
(2)Specificity=TN/(FP+TN)×100
(3)Accuracy=(TP+TN)/(TP+FP+FN+TN)×100
where *TP*, *FP*, *TN*, and *FN* represent true positives, false positives, true negatives, and false negatives, respectively. *TP* is the number of fall files in which the window is classified as a fall for the first time, while *FN* is the number of fall files in which all windows are inferred as a non-fall. *TN* represents the number of ADL files in which all windows are classified as a non-fall, while *FP* is the number of ADL files in which at least one window is misclassified as a fall. The window was classified as a fall or a non-fall based on a threshold of 0.7 for the SoftMax output [22]. Additionally, rapid detection of falls before an impact from the ground was also crucial in preimpact fall detection. As an indicator to represent it, the lead time was calculated as follows:(4)lead time=impact moment−detected moment 

## 3. Results

### 3.1. Performance of ConvLSTM

Figure 2 illustrates ConvLSTM models applied in this study. The original ConvLSTM model, ConvLSTM_9axis, was applied, and a variant, ConvLSTM_6axis, using only six-axis data (three-axis acceleration and three-axis angular velocity) as input data, was tested with (c). Additionally, the key concept of VGGNet shown in (a) was incorporated into the model, and a transition from the Conv1D-7 Block to the Conv1D-3 Block was made, as illustrated in (b), resulting in the creation of the ConvLSTM_6axis_VGG model. Table 4 represents the performance of ConvLSTM models. The model trained with nine-axis data demonstrated a sensitivity of 99.77%, a specificity of 95.27%, an accuracy of 97.37%, a lead time of 618.28 ± 331.38 ms, and a memory size of 1.58 MB. However, when the model was trained with only six-axis data, the accuracy (97.05%) and memory size (1.57 MB) slightly decreased. When the filter size was changed from seven to three, following the idea of VGGNet, the model became deeper, and the accuracy was almost restored. However, the memory size increased.

### 3.2. Performance of VGGNet

Table 5 shows the performance of VGGNet models. The original VGG16 model exhibited poor performance, with 49.55% sensitivity, 98.22% specificity, 75.50% accuracy, and 218.91 ± 173.85 ms lead time, despite its large memory size (56.6 MB). However, adding a batch normalization structure, similar to the ConvLSTM model, dramatically increased the accuracy to 97.27% while also increasing the lead time to 576.33 ± 271.26 ms. Furthermore, adding LSTM layers resulted in higher accuracy (98.32%) than the ConvLSTM benchmark model (97.37%), but with a larger memory size (58.7 MB). By setting the number of filters to 64, similar to ConvLSTM model, the memory usage (2.82 MB) was significantly reduced while maintaining almost the same accuracy (98.00%). However, further reduction in memory usage was necessary.

### 3.3. Performance of ResNet

Table 6 shows the performance of ResNet models. The ResNet50 model, with a bottleneck structure, which was the smallest version introduced in the paper [20], was implemented. The Original ResNet50 model demonstrated good performances, with a sensitivity of 99.32%, a specificity of 96.45%, an accuracy of 97.79%, and a lead time of 486.78 ± 241.08 ms. However, its memory size was too large. To address this issue, the number of filters was continuously reduced by half, resulting in a decrease in both accuracy and memory demand. A trade-off existed between the accuracy and the memory. When all the numbers of filters were set to 64, such as in ConvLSTM, a balance was achieved. Nevertheless, further reduction in memory usage was necessary, which made it challenging to demonstrate the advantage of ResNet.

### 3.4. Comparison of VGGNet and ResNet

Both VGGNet16 and ResNet50 were applied, but direct comparison was difficult due to their different structures. ResNet24 was created using VGGNet19 as the base to facilitate a comparison between VGGNet and ResNet. VGGNet19 was chosen because it has an even number of VGG Blocks. Every two VGG Blocks in VGGNet19 were replaced with Identity Blocks or Convolutional Blocks to create ResNet24, as shown in Figure 3. Despite requiring less memory than VGGNet19, higher accuracy was demonstrated by ResNet24 (Table 7), and it outperformed the benchmark model, ConvLSTM.

### 3.5. Lightweighted ResNet

The ResNet24 model developed in the previous section was modified to reduce memory demand. ResNet was composed of Identity Blocks and Convolutional Blocks. Convolutional Blocks abstract the data size, while Identity Blocks maintain the data size. Therefore, factors that did not affect the output size were removed one by one, as shown in Figure 4. As a result, the memory demand was continuously reduced without a dramatic change in other performances (Table 8). The compressed ResNet14 model has been named TinyFallNet as the final designation.

### 3.6. Evaluation Based on the Older Subject Dataset

The performance of TinyFallNet on elderly data was evaluated using both FARSEEING and KFall datasets. The FARSEEING dataset, which only contains fall data, was used for sensitivity evaluation, while the KFall dataset, with only ADL data, was employed for specificity evaluation. Since some parts of the FARSEEING dataset only have acceleration data, evaluation was carried out using a version of ResNet14 trained solely on three-axis acceleration data. Additionally, axis and unit conversions were performed to align with the KFall dataset. The evaluation showed a sensitivity of 80% (12/15), a specificity of 87.81% (562/640), and a lead time of 500.0 ± 0.0 ms. Table 9 shows the false detection in the KFall older adult dataset. If any trial resulted in a prediction failure, it was highlighted in red and bold text. All static actions were detected correctly, whereas numerous false detections occurred in movements that involved significant shifts in the center of gravity.

The current algorithm identifies a fall if at least one window was detected as a fall through window shifting. Elderly individuals tend to have a decreased sense of balance compared to young adults, which often results in windows being identified as falls within ADL actions, thereby reducing specificity. In order to address these issues, the algorithm was improved as follows. A fall was ultimately determined only when three consecutive windows were identified as a fall to prevent false detections in cases where balance was temporarily lost and then regained. The step size of the shifting window was changed from 10 to 1 in order to enable the model to make more frequent judgments and to ensure a sufficient window for capturing preimpact falls, which often occur within a short period of time. The application of the modified algorithm resulted in a sensitivity of 86.67% (13/15), a specificity of 97.97% (627/640), and a lead time of 477.7 ± 5.8 ms. Table 10 shows the false detection in the KFall older adult dataset using the modified algorithm. All static actions were detected correctly, whereas numerous false detections occurred in actions that involved significant shifts in the center of gravity. The number of false detections dramatically decreased.

## 4. Discussion

In this study, a preimpact fall-detection system was developed using lightweighting and outperforming deep learning models for wearable airbags. Our benchmark model was ConvLSTM [17,18,22] which is the state-of-the-art model in this field. ConvLSTM exhibited the best performance when applied to nine-axis data, which included three-axis accelerations, three-axis angular velocities, and three-axis Euler angles. The purpose of this study was to improve the ConvLSTM model by using only six-axis data, comprising three-axis accelerations and three-axis angular velocities, making it more lightweight and powerful.

The previous study [21] that applied AlexNet in post-fall detection demonstrated good performance. In this study, more improved models such as VGGNet and ResNet were attempted to be applied in preimpact fall detection, and theu showed good performance, as expected. It was observed that changing the filter size of ConvLSTM from seven to three, following the concept of VGGNet, resulted in an improvement in accuracy. Moreover, an improvement in accuracy was also observed when adding a BN layer and LSTM to ConvLSTM. This suggests that ConvLSTM and VGGNet have individual advantages and the potential to create synergy when combined.

The performance of VGGNet and ResNet was maintained even when their filter size was reduced because their original target data were images and have a larger input size than our preimpact fall detection data. In general, deep learning models designed for image processing require more filters. On the other hand, our target data were time-series and have a smaller input size than images. The optimal number of filters for their model was determined through trial and error in a previous study [17]. When the suggested number of filters, 64, was applied to VGGNet and ResNet, their memory demands were reduced without any decrease in performance. However, VGGNet and ResNet required some tuning to make them suitable for preimpact fall-detection applications.

ResNet24 was created by modifying VGG19 to compare the performance of the two models. ResNet24 demonstrated higher accuracy while requiring less memory than VGG19. This was achieved through some efforts, such as the bottleneck structure and global average pooling layer, to reduce memory usage and allow for deeper model architectures. The performance of ResNet in this comparison confirmed its superiority over VGGNet.

ResNet24 was made lightweight by removing one layer at a time, except for the factors that affect the output size. As the layers were continuously reduced, the memory demand decreased, but there was no significant change in other performance metrics. Since our target data were simpler than the image, TinyFallNet (ResNet14), with narrower layers, was sufficient for our needs.

When the TinyFallNet model proposed in this study was evaluated with data of older adults, it showed decreased performance, with a sensitivity of 80% (12/15) and a specificity of 88.43% (566/640). In the case of FARSEEING, it does not provide accurate situation information, making it difficult to identify the cause of false detections. The decreasing performance was presumed to be due to the fact that fall detection was attempted using only acceleration data. On the other hand, with KFall data, which consist of simulated ADL actions, the cause of the false detections could be inferred. The main cause of false detections appears to be the difference in movement patterns between older adults and young adults, as all static actions were accurately identified. Interestingly, most false detections occurred during actions with significant changes in the center of gravity and in participants who were unable to perform certain actions due to mobility limitations. This suggests that actions with significant shifts in the center of gravity carry a higher risk of falls, and elderly individuals with mobility issues are more susceptible to falls. Further analysis was challenging without video information, but these results underscore the importance of considering not only the balanced actions of young adults but also the unbalanced actions of older adults in developing a more robust fall detection algorithm. Therefore, some modifications were applied to develop a robust fall-detection algorithm considering the trends in data of older adults. This resulted in improved performance, with a sensitivity of 86.67% (13/15), a specificity of 97.97% (627/640), and a lead time of 477.7 ± 5.8 ms. ConvLSTM, a benchmark model, exhibited a sensitivity of 93.33%, a specificity of 99.84%, and a lead time of 411 ± 317 ms, when evaluated on the same dataset. For the KFall dataset, ConvLSTM utilized nine-axis data, enabling more precise pattern recognition and resulting in higher specificity. On the other hand, for the FARSEEING dataset, ConvLSTM utilized only three-axis accelerometer data and demonstrated better detection of individual fall events. The proposed algorithm showed a longer lead time, which could be attributed to the choice of setting the step size to one, requiring more frequent judgments.

This work has some limitations. Firstly, the model’s memory size is still too large for embedding in an MCU. Testing with a simple model implemented only with dense layers revealed that the required model size for embedding on an Arduino Nano 33 BLE Sense board, as supported by the TinyML group [26], was approximately 0.04 MB. Additional lightweighting is necessary for this purpose. It is believed that utilizing smaller models, such as MobileNet [27] or ShuffleNet [28] commonly used in image classification, and further tuning, can make embedding in an MCU feasible. Additionally, the use of knowledge-distillation techniques is believed to aid in further lightweighting [29]. The second limitation is that the sensitivity of our model decreased slightly compared to the benchmarked ConvLSTM model despite efforts to maintain performance. Generally, sensitivity is considered more crucial from a safety perspective. However, frequent false alarms can cause user discomfort and lead to users forgetting to wear the device [30]. Moreover, when applied to wearable airbags, false alarms leading to airbag deployment can become another cause of injury, especially for elderly individuals with impaired balance. Therefore, specificity is also important.

## 5. Conclusions

The hypotheses that image-based VGGNet and ResNet could be useful for preimpact fall detection and that the structures of these two models could contribute to the development of lightweight yet high-performance fall-detection algorithms were both correct. This study tried to apply CNN techniques used for images to preimpact fall detection and successfully achieved model lightweighting. The proposed TinyFallNet model exhibited higher accuracy (98.00%) and required less memory (0.70 MB) with six-axis data compared to the benchmark ConvLSTM model (accuracy: 97.37%; memory: 1.58 MB) with nine-axis data. Although additional research on model structure would be necessary to embed it into an MCU, this study was able to provide such a possibility.

## Figures and Tables

**Figure 1 sensors-23-08459-f001:**
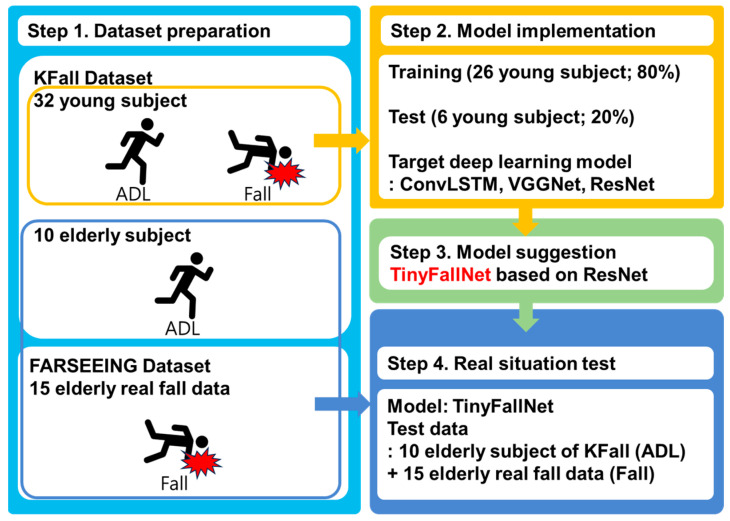
Overall flowchart of this study.

**Figure 2 sensors-23-08459-f002:**
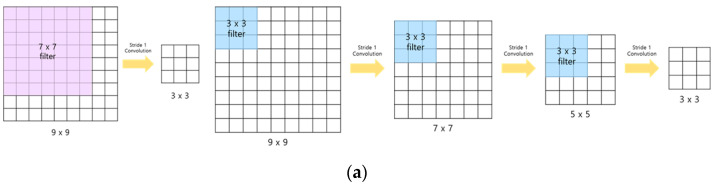
Structure of ConvLSTM models. (**a**) Key concept of VGGNet; (**b**) Conv1D-7 Block and Conv1D-3 Block; (**c**) ConvLSTM models.

**Figure 3 sensors-23-08459-f003:**
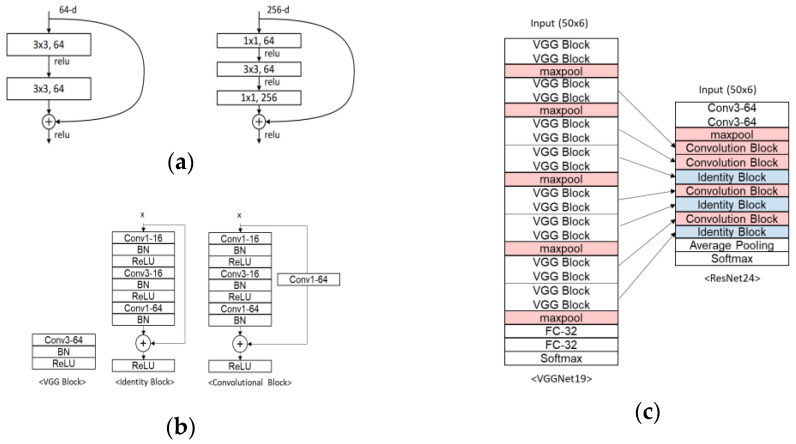
Structure of VGGNet19 and ResNet24. (**a**) Structure of the Bottleneck Layer; (**b**) VGG Block, Identity Block and Convolutional Block; (**c**) VGGNet19 and ResNet24.

**Figure 4 sensors-23-08459-f004:**
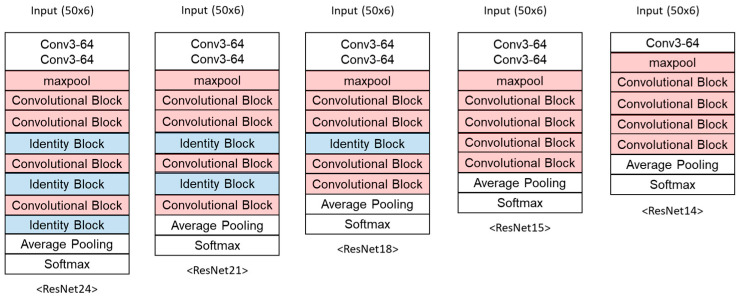
Structure of the lightweighted ResNet.

**Table 1 sensors-23-08459-t001:** ADLs in the KFall dataset.

Task ID	Activity	Trials
D01	Stand for 30 s	1
D02	Stand, slowly bend the back with or without bending at knees, tie shoelace, and get up	5
D03	Pick up an object from the floor	5
D04	Gently jump (try to reach an object)	5
D05	Stand, sit to the ground, wait a moment, and get up with normal speed	5
D06	Walk normally with turn (4 m)	5
D07	Walk quickly with turn (4 m)	5
D08	Jog normally with turn (4 m)	5
D09	Jog quickly with turn (4 m)	5
D10	Stumble while walking	5
D11	Sit on a chair for 30 s	1
D12	Sit on the sofa (back is inclined to the support) for 30 s	1
D13	Sit down to a chair normally, and get up from a chair normally	5
D14	Sit down to a chair quickly, and get up from a chair quickly	5
D15	Sit a moment, trying to get up, and collapse into a chair	5
D16	Stand, sit on the sofa (back is inclined to the support), and get up normally	5
D17	Lie on the bed for 30 s	1
D18	Sit a moment, lie down to the bed normally, and get up normally	5
D19	Sit a moment, lie down on the bed quickly, and get up quickly	5
D20	Walk upstairs and downstairs normally (five steps)	5
D21	Walk upstairs and downstairs quickly (five steps)	5

**Table 2 sensors-23-08459-t002:** Falls in KFall dataset.

Task ID	Activity	Trials
F01	Forward fall when trying to sit down	5
F02	Backward fall when trying to sit down	5
F03	Lateral fall when trying to sit down	5
F04	Forward fall when trying to get up	5
F05	Lateral fall when trying to get up	5
F06	Forward fall while sitting, caused by fainting	5
F07	Lateral fall while sitting, caused by fainting	5
F08	Backward fall while sitting, caused by fainting	5
F09	Vertical (forward) fall while walking caused by fainting	5
F10	Fall while walking, use of hands to dampen fall, caused by fainting	5
F11	Forward fall while walking caused by a trip	5
F12	Forward fall while jogging caused by a trip	5
F13	Forward fall while walking caused by a slip	5
F14	Lateral fall while walking caused by a slip	5
F15	Backward fall while walking caused by a slip	5

**Table 3 sensors-23-08459-t003:** Falls in FARSEEING dataset.

Task ID	Activity
17744725-01	After walking with a wheeled walker, stood behind a chair, then fell backward on the floor.
38243026-05	Fell forward while bending down to fix the shoelace.
42990421-01	Wanted to pick up an object from the ground
42990421-02	Got up from the chair and wanted to walk
42990421-03a	When trying to move to the side, wheeled walker fell forward. Freezed movement.
72858619-01	Went to the table in the dining room. Fell down backward on the buttocks.
72858619-02	Person held onto the wall, then fell down backward on the buttocks.
74827807-04	Fell while walking.
74827807-07	Walked, then fell down in front of the entrance of the house.
79761947-03	While changing the hip protector, fell backward on the ground, and hit the toilet.
91943076-01	Fell down in front of the entrance of the house.
91943076-02	Walked, then fell down opening the door in the entrance hall.
96201346-01	While walking to the bathroom, stopped by freezing, and fell from standing position.
96201346-03	Standing at the wardrobe, wanted to walk backward and then fell on buttocks.
96201346-05	Walked backward from the washing basin, then fell backward.

**Table 4 sensors-23-08459-t004:** Classification performances of ConvLSTM models.

	ConvLSTM_9axis	ConvLSTM_6axis	ConvLSTM_6axis_VGG
Sensitivity (%)	99.77	100.00	99.77
Specificity (%)	95.27	94.48	95.07
Accuracy (%)	97.37	97.05	97.27
Lead time (ms)	618.28 ± 331.38	558.76 ± 253.81	548.60 ± 289.10
Memory size (MB)	1.58	1.57	2.07

**Table 5 sensors-23-08459-t005:** Classification performances of VGGNet models.

	VGG16	VGG16 + BN	VGG16 + BN + LSTM	VGG16 + BN + LSTM + (Filters = 64)
Sensitivity (%)	49.55	100.00	99.55	100.00
Specificity (%)	98.22	94.87	97.24	96.25
Accuracy (%)	75.50	97.27	98.32	98.00
Lead time (ms)	218.91 ± 173.85	576.33 ± 271.26	545.77 ± 201.17	560.56 ± 312.31
Memory size (MB)	56.6	56.9	58.7	2.82

**Table 6 sensors-23-08459-t006:** Classification performances of ResNet models.

	ResNet50	ResNet50 (Filters 1/2)	ResNet50 (Filters 1/4)	ResNet50 (Filters 1/8)	ResNet50 (Filters 1/16)	ResNet50 (Filters 1/32)	ResNet50 (Filters= 64)
Sensitivity (%)	99.32	98.65	97.52	98.20	89.86	59.23	97.30
Specificity (%)	96.45	96.84	96.84	95.27	97.04	97.44	96.84
Accuracy (%)	97.79	97.69	97.16	96.64	93.69	79.60	97.06
Lead time (ms)	486.78 ± 241.08	477.03 ± 288.65	463.56 ± 261.34	498.30 ± 312.57	318.70 ± 194.31	276.16 ± 159.09	445.16 ± 243.49
Memory size (MB)	184.0	47.0	12.6	3.95	1.74	1.17	5.07

**Table 7 sensors-23-08459-t007:** Classification performances of VGGNet19 and ResNet24.

	VGG19	ResNet24
Sensitivity (%)	99.09	99.55
Specificity (%)	95.86	97.63
Accuracy (%)	97.37	98.5
Lead time (ms)	583.20 ± 336.42	561.72 ± 309.12
Memory size (MB)	2.68	1.11

**Table 8 sensors-23-08459-t008:** Classification performances of lightweighted ResNet models.

	ResNet24	ResNet21	ResNet18	ResNet15	ResNet14
Sensitivity (%)	99.55	100.00	99.77	98.65	99.32
Specificity (%)	97.63	95.66	96.25	97.63	96.84
Accuracy (%)	98.53	97.69	97.90	98.11	98.00
Lead time (ms)	561.72 ± 309.12	558.31 ± 302.86	495.30 ± 230.88	493.54 ± 274.58	539.00 ± 278.07
Memory size (MB)	1.11	1.01	0.96	0.87	0.70

**Table 9 sensors-23-08459-t009:** False detection in the KFall older adult dataset using the TinyFallNet.

Movement	Subject									
	S1	S2	S3	S4	S5	S6	S7	S8	S9	S10
D01	1/1	1/1	1/1	1/1	1/1	1/1	1/1	1/1	1/1	1/1
D02	** 0/5 **	5/5	5/5	5/5	** 4/5 **	NA	5/5	5/5	5/5	5/5
D03	** 3/5 **	5/5	5/5	5/5	5/5	NA	5/5	** 4/5 **	** 3/5 **	** 3/5 **
D04	NA	NA	NA	NA	NA	NA	NA	NA	NA	NA
D05	5/5	** 2/5 **	NA	** 2/5 **	** 1/5 **	NA	NA	NA	NA	5/5
D06	5/5	5/5	** 4/5 **	5/5	5/5	** 0/5 **	** 4/5 **	5/5	5/5	5/5
D07	5/5	5/5	5/5	5/5	5/5	** 0/5 **	5/5	5/5	5/5	5/5
D08	5/5	5/5	5/5	5/5	5/5	5/5	5/5	5/5	5/5	5/5
D09	5/5	5/5	NA	5/5	5/5	NA	5/5	5/5	5/5	5/5
D10	NA	NA	NA	NA	NA	NA	NA	NA	NA	NA
D11	1/1	1/1	1/1	1/1	1/1	1/1	1/1	1/1	1/1	1/1
D12	1/1	1/1	1/1	1/1	1/1	1/1	1/1	1/1	1/1	1/1
D13	5/5	5/5	5/5	5/5	5/5	5/5	5/5	5/5	5/5	** 4/5 **
D14	5/5	5/5	** 4/5 **	5/5	5/5	** 1/5 **	** 4/5 **	5/5	5/5	** 4/5 **
D15	NA	NA	NA	NA	NA	NA	NA	NA	NA	NA
D16	5/5	5/5	5/5	5/5	5/5	** 4/5 **	** 4/5 **	5/5	5/5	5/5
D17	1/1	1/1	1/1	1/1	1/1	1/1	1/1	1/1	1/1	1/1
D18	5/5	5/5	5/5	5/5	** 3/5 **	5/5	5/5	5/5	5/5	5/5
D19	5/5	5/5	5/5	5/5	5/5	5/5	5/5	5/5	5/5	** 4/5 **
D20	** 4/5 **	** 1/5 **	** 0/5 **	** 1/5 **	** 0/5 **	** 0/5 **	NA	** 0/5 **	** 4/5 **	5/5
D21	NA	NA	NA	NA	NA	NA	NA	NA	NA	NA

The fractions denote the number of correctly predicted data/total number of the data.

**Table 10 sensors-23-08459-t010:** False detection in the KFall older adult dataset using the modified TinyFallNet.

Movement	Subject									
	S1	S2	S3	S4	S5	S6	S7	S8	S9	S10
D01	1/1	1/1	1/1	1/1	1/1	1/1	1/1	1/1	1/1	1/1
D02	5/5	5/5	5/5	5/5	5/5	NA	5/5	5/5	5/5	5/5
D03	5/5	5/5	5/5	5/5	5/5	NA	5/5	** 4/5 **	5/5	** 4/5 **
D04	NA	NA	NA	NA	NA	NA	NA	NA	NA	NA
D05	5/5	5/5	NA	5/5	** 3/5 **	NA	NA	NA	NA	5/5
D06	5/5	5/5	5/5	5/5	5/5	5/5	5/5	5/5	5/5	5/5
D07	5/5	5/5	5/5	5/5	5/5	5/5	5/5	5/5	5/5	5/5
D08	5/5	5/5	5/5	5/5	5/5	5/5	5/5	5/5	5/5	5/5
D09	5/5	5/5	NA	5/5	5/5	NA	5/5	5/5	5/5	5/5
D10	NA	NA	NA	NA	NA	NA	NA	NA	NA	NA
D11	1/1	1/1	1/1	1/1	1/1	1/1	1/1	1/1	1/1	1/1
D12	1/1	1/1	1/1	1/1	1/1	1/1	1/1	1/1	1/1	1/1
D13	5/5	5/5	5/5	5/5	5/5	5/5	5/5	5/5	5/5	5/5
D14	5/5	5/5	5/5	5/5	5/5	5/5	5/5	5/5	5/5	** 4/5 **
D15	NA	NA	NA	NA	NA	NA	NA	NA	NA	NA
D16	5/5	5/5	5/5	5/5	5/5	5/5	5/5	5/5	5/5	5/5
D17	1/1	1/1	1/1	1/1	1/1	1/1	1/1	1/1	1/1	1/1
D18	5/5	5/5	5/5	5/5	5/5	5/5	5/5	5/5	5/5	5/5
D19	5/5	5/5	5/5	5/5	5/5	5/5	5/5	5/5	5/5	5/5
D20	5/5	5/5	** 0/5 **	5/5	5/5	** 2/5 **	NA	5/5	5/5	5/5
D21	NA	NA	NA	NA	NA	NA	NA	NA	NA	NA

The fractions denote the number of correctly predicted data/total number of the data.

## Data Availability

No new data were created.

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
