# Peer review of "TinyFallNet: A Lightweight Pre-Impact Fall Detection Model"

_sensors, 2023, doi:10.3390/s23208459_

Round 1

Reviewer 1 Report

I find this paper interesting and thank you to all authors. Could you kindly elucidate the primary contributions made by the authors in this paper? Is the main emphasis of the paper on comparing the performance of different deep learning algorithms across various datasets? Additionally, since you mentioned the dataset collected by Yu et al., could you provide insights into their study (in revised version) as well?

Author Response

We appreciate your interest in our paper and providing helpful comments. we have made the following revisions based on the comments you provided.

<Response 1>

A new paragraph was added to express the contributions and emphasis in lines 92~111. We explained the aim of our study and its contributions.

<Revised manuscript 1>

In this work, we present to lightweight a pre-impact fall detection algorithm using deep learning models commonly employed for image classification. ConvLSTM, a state-of-the-art model for pre-impact fall detection, was benchmarked and attempts were made to lighten it while maintaining performance. In the pursuit of lightweight the model, VGGNet and ResNet, commonly used image classification models, were applied and analyzed for the pre-impact fall detection problem. As a result, TinyFallNet, based on ResNet, was proposed. The principal contributions of the present work are as follows: Firstly, we demonstrated the applicability of models developed for image classification to the IMU-based pre-impact fall detection problem. Secondly, we demonstrated additional tuning possibilities for image classification models. VGGNet and ResNet required fewer parameters in our problem than the image problem due to the difference in the data type. Especially, ResNet showed that performance was maintained despite removing the Identity Block. Thirdly, ConvLSTM model was successfully lightweight while maintaining performance. The lightweight model was named TinyFallNet and showed higher accuracy (98.00%) and required smaller memory (0.70 MB) compared to the benchmark ConvLSTM model (accuracy: 97.37%; memory: 1.58MB). Finally, the proposed TinyFallNet applied well to real-fall situations. The Farseeing dataset was employed to evaluate actual fall movements in the elderly, and the extended KFall dataset was used to assess the ADLs of the elderly. The results yielded a sensitivity of 86.67%, a specificity of 97.97%, and a lead time of 477.7±5.8 ms. (Line 92~111)

<Response 2>

The KFall dataset initially collected daily activities and fall movements from 32 young adults. Subsequently, data from 10 elderly individuals were added, focusing on both falls and expanding some daily activities. One significant reason for utilizing this dataset in our study is its role in the development of the benchmark model, ConvLSTM. Another key aspect is that it is the only dataset providing information on the onset and offset moments of falls. You can find further details on these aspects in the subsequent lines of the revision.

<Revised manuscript 2>

Yu et al., [16] developed a new dataset called "KFall" for pre-impact fall detection. This new dataset consisted of 3-axis accelerations, 3-axis angular velocities, and 3-axis Euler angles, which were collected from 32 Korean participants performing 21 ADLs and 15 falls. They provided the fall onset moment by integrating information from sensor and video data. (Line 63~67)

The extended KFall dataset [20] provides ADL data from the older adults (Line 89~90)

The KFall dataset [16,20] was used in the study for two reasons: firstly, it was the first public dataset suitable for pre-impact fall detection due to provide the time information about the onset and offset of the fall; and secondly, ConvLSTM, which served as our benchmark model, was developed using KFall. (Line 123~125)

Reviewer 2 Report

Dear Authors,

I think your paper “TinyFallNet: A lightweight pre-impact fall detection model” is interesting to read and presents a possible solution to a very important problem of fall detection and harm prevention. The article presents a pre-fall detection system based on machine learning that demonstrates good performance and accuracy. The study uses a public KFall dataset and three deep learning models (ConvLSTM, 12 VGGNet, and ResNet), which achieve more than 98% accuracy and a very small memory footprint. The paper is quite well written, presenting important findings and results. Despite that, I have several comments and suggestions which would allow the authors to improve the paper even more:

1. The Abstract section should be slightly expanded to reflect the Background (contain the problem addressed in a broad context and highlight the purpose of the study) and Conclusions (indicate the main conclusions or interpretations) as required by the Sensors template. Because at this moment abstract presents only results.

2. Acronyms/Abbreviations/Initialisms should be defined the first time they appear in each of three sections: the abstract; the main text; the first figure or table. When defined for the first time, the acronym/abbreviation/initialism should be added in parentheses after the written-out form. For example, what is the meaning of IMU, MCU, CNN, ADL, etc.? Some readers may not be familiar with these terms.

3. The Materials and Methods section must be improved because it is not clear which subjects and data were used at the different stages of model development and testing:

3.1. Lines 101-102 say that the dataset was collected from 32 young males and 10 old subjects (42 in total). Then lines 108-111 say: “The dataset also included 21 daily living (ADLs) and 15 simulated falls (Table 1 and Table 2). Young subjects were required to perform both ADLs and falls, while old subjects did not perform any falls and several ADLs involving risk, including D04, D10, D15 and D21.”

3.1.1. So, it is unclear how many records of each type does the dataset contain? What does the column “Trials” in the tables mean?

3.2. Later, the 172-172 lines say that for model training 26 subjects’ data were used, while 6 subjects’ data were used for testing (total 32??).

3.2.1 It is unclear if these are the same subjects mentioned in lines 101-102? If so, what was the total number of subjects? Because in different places you provide contradictory data (42 subjects vs. 32 subjects). Again, what was the size of the dataset? Is this the same KFall dataset?

4. To avoid such confusion regarding your methodology and datasets, I strongly recommend that you include some kind of flowchart (activity diagram) at the beginning of the Materials and Methods section that represents the main steps of your methodology and describes each step in detail.

5. Please add legends to Tables 9 and 10 or explain them in the text, since it is difficult to understand what data is presented in these tables (what do the fractions mean?).

6. There are many references older than 5 years (more than half). I believe that most of them can be replaced with newer ones, reflecting the current state of the art.

The quality of the English language is good. I found only minor spelling errors and typos, which can be easily corrected with spellcheckers.

Author Response

Dear Authors,

I think your paper “TinyFallNet: A lightweight pre-impact fall detection model” is interesting to read and presents a possible solution to a very important problem of fall detection and harm prevention. The article presents a pre-fall detection system based on machine learning that demonstrates good performance and accuracy. The study uses a public KFall dataset and three deep learning models (ConvLSTM, 12 VGGNet, and ResNet), which achieve more than 98% accuracy and a very small memory footprint. The paper is quite well written, presenting important findings and results. Despite that, I have several comments and suggestions which would allow the authors to improve the paper even more:

We appreciate your interest in my paper and the valuable feedback you provided. Following your comments, we have made the following revisions.

  1. The Abstract section should be slightly expanded to reflect the Background (contain the problem addressed in a broad context and highlight the purpose of the study) and Conclusions (indicate the main conclusions or interpretations) as required by the Sensors template. Because at this moment abstract presents only results.

<Response 1>

Thank you for pointing out our mistake. Based on your comments, we have revised the abstract as follows.

<Revised manuscript 1>

Abstract: Falls represent a significant health concern for the elderly. While studies on deep learning-based pre-impact fall detection have been conducted to mitigate fall-related injuries, additional efforts are needed for embedding in microcomputer units (MCUs). In this study, ConvLSTM, the state-of-the-art model, was benchmarked and was attempted to lightweight it by leveraging features from image classification models VGGNet and ResNet while maintaining performance. The models were developed and evaluated using data from young subjects in the KFall public dataset based on inertial measurement unit (IMU), leading to the proposal of TinyFallNet based on ResNet. Despite exhibiting higher accuracy (97.37% < 98.00%) than the benchmarked ConvLSTM, the proposed model requires lower memory (1.58 MB > 0.70 MB). Additionally, elderly data from fall data of FARSEEING dataset and activities of daily living (ADLs) data of KFall dataset were analyzed for algorithm validation. This study demonstrated the applicability of image classification models to pre-impact fall detection using IMU and showed that additional tuning for lightweighting is possible due to the different data types. This research was expected to contribute to the lightweighting of deep learning models based on IMU and the development of applications based on IMU data. (Line 11~24)

  1. Acronyms/Abbreviations/Initialisms should be defined the first time they appear in each of three sections: the abstract; the main text; the first figure or table. When defined for the first time, the acronym/abbreviation/initialism should be added in parentheses after the written-out form. For example, what is the meaning of IMU, MCU, CNN, ADL, etc.? Some readers may not be familiar with these terms.

<Response 2>

Thank you for good suggestions for our paper. We have revised the abstract as follows based on your comments.

<Revised manuscript 2>

While studies on deep learning-based pre-impact fall detection have been conducted to mitigate fall-related injuries, additional efforts are needed for embedding in microcomputer units (MCUs) (Line 11~13)

The models were developed and evaluated using data from young subjects in the KFall public dataset based on inertial measurement unit (IMU), leading to the proposal of TinyFallNet based on ResNet. (Line 15~17)

Additionally, elderly data from fall data of FARSEEING dataset and activities of daily living (ADLs) data of KFall dataset were analyzed for algorithm validation. (Line 19~21)

Convolutional Neural Network (CNN)-based classifiers have been primarily used for image processing, so attention was given to models used in image classification. (Line 77~79)

  1. The Materials and Methods section must be improved because it is not clear which subjects and data were used at the different stages of model development and testing:

3.1. Lines 101-102 say that the dataset was collected from 32 young males and 10 old subjects (42 in total). Then lines 108-111 say: “The dataset also included 21 daily living (ADLs) and 15 simulated falls (Table 1 and Table 2). Young subjects were required to perform both ADLs and falls, while old subjects did not perform any falls and several ADLs involving risk, including D04, D10, D15 and D21.”

3.1.1. So, it is unclear how many records of each type does the dataset contain? What does the column “Trials” in the tables mean?

<Response 3.1.1>

Each activity was repeated for a specific number of trials. The term "trials" denotes the number of repetitions for each activity, following the notation used in the referenced paper. This addition is made to enhance the understanding of our readers.

<Revised manuscript 3.1.1>

Each activity was performed for a repeated number of trials. (Line 134)

3.2. Later, the 172-172 lines say that for model training 26 subjects’ data were used, while 6 subjects’ data were used for testing (total 32??).

<Response 3.2>

We apologize for the confusion. Additional clarifications have been included as follows to provide clarity.

<Revised manuscript 3.2>

A sufficient amount of data patterns is essential to train the deep learning model. Therefore, only data from young participants (32 subjects) in the KFall dataset were used for model development. Data from elderly individuals (10 subjects) in KFall were reserved solely for the evaluation of the finally proposed model. (Line 199~203)

3.2.1 It is unclear if these are the same subjects mentioned in lines 101-102? If so, what was the total number of subjects? Because in different places you provide contradictory data (42 subjects vs. 32 subjects). Again, what was the size of the dataset? Is this the same KFall dataset?

<Response 3.2.1>

We apologize once again for the confusion. To provide further clarification, the data from elderly individuals in KFall consists only of Activities of Daily Living (ADL) actions, and the quantity of this data is not sufficient for developing a deep learning model. Therefore, it was deemed unsuitable for the development of the model and was solely utilized for the final evaluation of the proposed model. The previously added sentence above is intended to assist in better understanding this context.

<Revised manuscript 3.2.1>

A sufficient amount of data patterns is essential to train the deep learning model. Therefore, only data from young participants (32 subjects) in the KFall dataset were used for model development. Data from elderly individuals (10 subjects) in KFall were reserved solely for the evaluation of the finally proposed model. (Line 199~203)

  1. To avoid such confusion regarding your methodology and datasets, I strongly recommend that you include some kind of flowchart (activity diagram) at the beginning of the Materials and Methods section that represents the main steps of your methodology and describes each step in detail.

<Response 4>

As per your suggestion, I have incorporated the overall flowchart as follows.

<Revised manuscript 3.4>

2.1. Overall flowchart

Figure 1 illustrates the overall flowchart of this study. Firstly, the dataset for model development and evaluation was prepared. In the next step, a model was implemented using only the data from young participants in the KFall dataset [16], and efforts were made to lightweight the model. Based on this, TinyFallNet was proposed. Finally, the model was evaluated using the elderly ADL data from the KFall dataset [20] and 15 actual fall datasets from the FARSEEING dataset [21]. (Line 114~119)

Figure 1. overall flowchart of this study (line 120~121)

  1. Please add legends to Tables 9 and 10 or explain them in the text, since it is difficult to understand what data is presented in these tables (what do the fractions mean?).

<Response 5>

According to your comment, following legend was added in Table 9 and 10.

<Revised manuscript 5>

※ The fractions denoted ‘the number of correctly predicted data/total number of the data’ (Table 9 and 10)

In addition, we added the following sentence to aid understanding.

If any trial resulted in a prediction failure, it was highlighted in red text. (Line 297~298)

  1. There are many references older than 5 years (more than half). I believe that most of them can be replaced with newer ones, reflecting the current state of the art.

<Response 6>

In citing the papers for explaining the background of the research, it was unavoidable to select older papers as they were the initial references [1-12]. Additionally, the models used for image classification may seem relatively unfamiliar in the inertial sensor domain, but they have been established for a longer time in the field of image processing [17,18]. However, the key papers that are directly relevant to my research are mostly recent publications, as I believe [13-16,19-21,23]. Since replacing the cited papers with new ones is considered difficult, I have added a limitation section and included references to more recent papers for a more up-to-date perspective [24-27]. I hope this response meets your expectations.

<Revised manuscript 6>

This work has some limitations. Firstly, the model's memory size is still too large for embedding in an MCU. Testing with a simple model implemented only with dense layers revealed that the required model size for embedding on an Arduino Nano 33 BLE Sense board, as supported by the TinyML group [24], was approximately 0.04MB. Additional lightweight is necessary for this purpose. It is believed that utilizing smaller-sized models, such as MobileNet [25] or ShuffleNet [26] commonly used in image classification, and further tuning, can make embedding in an MCU feasible. Additionally, the use of knowledge distillation techniques is believed to aid in further lightweight [27]. The second limitation is that the sensitivity of our model decreased slightly compared to the benchmarked ConvLSTM model despite efforts to maintain performance. Generally, sensitivity is considered more crucial from a safety perspective. However, frequent false alarms can cause user discomfort and lead to users forgetting to wear the device [28]. Moreover, when applied to wearable airbags, false alarms leading to airbag deployment can become another cause of injury, especially for elderly individuals with impaired balance. Therefore, the specificity is also important. (Line 379~393)

  1. Warden, P.; Situnayake, D. Tinyml: Machine Learning with Tensorflow Lite on Arduino and Ultra-Low-Power Microcontrollers; O’Reilly Media, 2019; ISBN 1492052019.
  2. Howard, A.; Sandler, M.; Chu, G.; Chen, L.-C.; Chen, B.; Tan, M.; Wang, W.; Zhu, Y.; Pang, R.; Vasudevan, V. Searching for Mobilenetv3. In Proceedings of the Proceedings of the IEEE/CVF international conference on computer vision; 2019; pp. 1314–1324.
  3. Zhang, X.; Zhou, X.; Lin, M.; Sun, J. ShuffleNet: An Extremely Efficient Convolutional Neural Network for Mobile Devices. In Proceedings of the Proceedings of the IEEE Computer Society Conference on Computer Vision and Pattern Recognition; 2018; pp. 6848–6856.
  4. Chi, T.; Liu, K.; Hsieh, C.; Tsao, Y.; Chan, C. PREFALLKD : PRE-IMPACT FALL DETECTION VIA CNN-VIT KNOWLEDGE DISTILLATION. ICASSP 2023 - 2023 IEEE Int. Conf. Acoust. Speech Signal Process. 1–5, doi:10.1109/ICASSP49357.2023.10094979.
  5. Mercuri, M.; Soh, P.J.; Mehrjouseresht, P.; Crupi, F.; Schreurs, D. Biomedical Radar System for Real-Time Contactless Fall Detection and Indoor Localization. IEEE J. Electromagn. RF Microwaves Med. Biol. 2023, PP, 1–10, doi:10.1109/JERM.2023.3278473.

Reviewer 3 Report

The overall novelty of the manuscript is very limited in my opinion. The authors applied open-sourced DL networks on an open-sourced fall detection dataset and drew the conclusion from the training/evaluation results. The techniques mentioned in the manuscript have been widely used in previous works of a similar topic and should not be considered novel. I would recommend the authors either implement the algorithm proposed on a real MCU or use methods that have not been performed on the fall detection task.

1.     In abstract line 23, I am assuming that Resnet14 should be renamed as resnet24

2.     The networks should be better visualized, including ConvLSTM and its variations

3.     The motivation of the manuscript is not well addressed: if the focus of the manuscript is to develop a lightweight network, why not consider a lightweight backbone such as MobileNet, ShuffleNet etc.. VGG is not suitable for MCU at all because of its large size and inefficient computation.

4.      Please address how the lead time is measured, 500+ms on an RTX3080 seems unreasonable to me.

5.     The Fall Detection task, as a safety-related feature, focuses more on sensitivity than on specificity. The author should demonstrate the supremacy of the proposed method in its sensitivity on the task. 

Presentation of the manuscript is satisfactory 

Author Response

I appreciate your thoughtful consideration of my paper and the valuable insights you provided from your expertise. Lightweighting of models has been extensively researched in the field of image processing for a long time. However, in the domain of inertial sensor-based fall detection, research has only begun in recent years due to a lack of datasets and technological advancements. From this perspective, I aimed to investigate whether models commonly used in image processing could be applied to our field and, in doing so, sought to lightweight the ConvLSTM model developed by my colleague Yu. Based on your valuable feedback, I have revised the paper as follows.

  1. In abstract line 23, I am assuming that Resnet14 should be renamed as resnet24

<Response 1>

I apologize for the confusion caused. The TinyFallNet that I proposed indeed refers to ResNet14. I inadvertently omitted additional clarification, leading to the confusion. The sentence has been revised as follows.

<Revised manuscript 1>

The models were developed and evaluated using data from young subjects in the KFall public dataset based on inertial measurement unit (IMU), leading to the proposal of TinyFallNet based on ResNet. (Line 15~17)

  1. The networks should be better visualized, including ConvLSTM and its variations

<Response 2>

Thank you for the valuable input. Following the comments you provided, I have visualized ConvLSTM and its variables as follows. Please see the attachment for the Figure 2.

<Revised manuscript 2>

Figure 2 illustrates the ConvLSTM models applied in this study. The original ConvLSTM model, ConvLSTM_9axis, was applied, and a variant, ConvLSTM_6axis, using only 6-axis data (3-axis acceleration and 3-axis angular velocity) as input data, was experimented with (c). Additionally, the key concept of VGGNet shown in (a) was incorporated into the model, and a transition from Conv1D-7 Block to Conv1D-3 Block was made, as illustrated in (b), resulting in the creation of the ConvLSTM_6axis_VGG model. (Line 225~230)

  1. The motivation of the manuscript is not well addressed: if the focus of the manuscript is to develop a lightweight network, why not consider a lightweight backbone such as MobileNet, ShuffleNet etc.. VGG is not suitable for MCU at all because of its large size and inefficient computation.

<Response 3.1>

Thank you for the positive comment. In order to provide a clearer explanation of the motivation behind this study, I have added the following content to the introduction.

<Revised manuscript 3.1>

In this work, we present to lightweight a pre-impact fall detection algorithm using deep learning models commonly employed for image classification. ConvLSTM, a state-of-the-art model for pre-impact fall detection, was benchmarked and attempts were made to lighten it while maintaining performance. In the pursuit of lightweight the model, VGGNet and ResNet, commonly used image classification models, were applied and analyzed for the pre-impact fall detection problem. As a result, TinyFallNet, based on ResNet, was proposed. (Line 93~98)

<Response 3.2>

This study originated from the curiosity about the potential outcomes if the Conv1D block with a kernel size of 7, used by ConvLSTM, were replaced with multiple Conv1D blocks with a kernel size of 3, similar to VGGNet. Consequently, the analysis focused on VGGNet and the more advanced ResNet. I appreciate your recommendation of the MobileNet and ShuffleNet models. Upon further investigation prompted by your comment, it appears that these models are designed for embedding in smartphones rather than MCUs. However, given that VGGNet and ResNet in this study have been lightened through tuning, I believe they are suitable for embedding inertial sensor-based fall detection models. I have mentioned this as a limitation and will strive to showcase promising research in the future.

<Revised manuscript 3.2>

This work has some limitations. Firstly, the model's memory size is still too large for embedding in an MCU. Testing with a simple model implemented only with dense layers revealed that the required model size for embedding on an Arduino Nano 33 BLE Sense board, as supported by the TinyML group [24], was approximately 0.04MB. Additional lightweight is necessary for this purpose. It is believed that utilizing smaller-sized models, such as MobileNet [25] or ShuffleNet [26] commonly used in image classification, and further tuning, can make embedding in an MCU feasible. Additionally, the use of knowledge distillation techniques is believed to aid in further lightweight [27]. (Line 379~386)

  1. Please address how the lead time is measured, 500+ms on an RTX3080 seems unreasonable to me.

<Response 4>

I have added the computation formula for lead time. This value is solely derived from the algorithm and is independent of the computing capabilities of the PC. In the future, when applied to an MCU, the computation time of the MCU may indeed have an impact.

<Revised manuscript 4>

Additionally, rapid detection of falls before an impact from the ground was also crucial in pre-impact fall detection. As an indicator to represent it, the lead time was calculated as follows: (Line 219~221)

lead time = impact moment - detected moment (4)

  1. The Fall Detection task, as a safety-related feature, focuses more on sensitivity than on specificity. The author should demonstrate the supremacy of the proposed method in its sensitivity on the task.

<Response 5>

Thank you for the valuable insight. As you pointed out, sensitivity is generally considered more important than specificity. However, the ultimate goal after embedding the model in an MCU is to apply it to wearable airbags. Frequent false alarms in wearable airbags can cause user discomfort, and elderly individuals with impaired balance might be at risk of injury from incorrectly inflated airbags. I have incorporated this information into the limitations section as follows.

<Revised manuscript 5

The second limitation is that the sensitivity of our model decreased slightly compared to the benchmarked ConvLSTM model despite efforts to maintain performance. Generally, the sensitivity is considered more crucial from a safety perspective. However, frequent false alarms can cause user discomfort and lead to users forgetting to wear the device [28]. Moreover, when applied to wearable airbags, false alarms leading to airbag deployment can become another cause of injury, especially for elderly individuals with impaired balance. Therefore, the specificity is also important. (Line 386~393)

Round 2

Reviewer 2 Report

Dear Authors,

Thank you for addressing all comments and suggestions. I have no any further comments and believe, that article can be published as it is.

Good luck in the future research!

Quality of English language is very high, article is well written (all minor errors can be corrected by academic editor).

Author Response

Thank you for your timely input and valuable advice to enhance the quality of our paper. Have a great day!

Reviewer 3 Report

1. The author should constrain the scope of the algorithm to wearable airbag application in title, abstract and etc, to further strengthen the novelty. 

2. I would doubt that the author claims that "in the domain of inertial sensor-based fall detection, research has only begun in recent years due to a lack of datasets and technological advancements". In fact, the topic has been researched for a long time, using with deep learning, and conventional method, for instance:

Syed, A.S.; Sierra-Sosa, D.; Kumar, A.; Elmaghraby, A. A Deep Convolutional Neural Network-XGB for Direction and Severity Aware Fall Detection and Activity Recognition. Sensors 202222, 2547. https://doi.org/10.3390/s22072547

Y. Iguchi, J. H. Lee and S. Okamoto, "Enhancement of Fall Detection Algorithm Using Convolutional Autoencoder and Personalized Threshold," 2021 IEEE International Conference on Consumer Electronics (ICCE), Las Vegas, NV, USA, 2021, pp. 1-5, doi: 10.1109/ICCE50685.2021.9427732.

F. Cruciani et al., "Comparing CNN and Human Crafted Features for Human Activity Recognition," 2019 IEEE SmartWorld, Ubiquitous Intelligence & Computing, Advanced & Trusted Computing, Scalable Computing & Communications, Cloud & Big Data Computing, Internet of People and Smart City Innovation (SmartWorld/SCALCOM/UIC/ATC/CBDCom/IOP/SCI), Leicester, UK, 2019, pp. 960-967, doi: 10.1109/SmartWorld-UIC-ATC-SCALCOM-IOP-SCI.2019.00190.

And the similar method was also used in processing temporal data not limited to IMU signals, for example:

Z. Ye, Y. Li, R. Jin and Q. Li, "Towards Accurate Odor Identification and Effective Feature Learning With an AI-Empowered Electronic Nose," in IEEE Internet of Things Journal, doi: 10.1109/JIOT.2023.3299555.

Thus I would recommend the authors to give a more detailed review on the related works

The presentation of the work is good in my opinion. 

Author Response

Thank you for your interest in our paper and for providing valuable advice to help improve it. Here is our responses to your kind review:

  1. The author should constrain the scope of the algorithm to wearable airbag application in title, abstract and etc, to further strengthen the novelty. 

<Response 1>

We appreciate your suggestion regarding the wording related to wearable airbags. Unfortunately, we have not succeeded in embedding the algorithm in an MCU for wearable airbag applications yet, and thus we are concerned about using the term "wearable airbag" in the title. However, we have added explanations in the abstract, introduction, and discussion sections to clarify the research purpose.

<Revised manuscript 1>

In this study, ConvLSTM, the state-of-the-art model, was benchmarked and was attempted to lightweight it by leveraging features from image classification models VGGNet and ResNet while maintaining performance for wearable airbag. (Line 13~16; Abstract)

In this work, we present to lightweight a pre-impact fall detection algorithm using deep learning models commonly employed for image classification for wearable airbag. (Line 106~107; Introduction)

In this study, a pre-impact fall detection system was developed using lightweight and outperforming deep learning models for wearable airbag. (Line 330~331; Discussion)

  1. I would doubt that the author claims that "in the domain of inertial sensor-based fall detection, research has only begun in recent years due to a lack of datasets and technological advancements". In fact, the topic has been researched for a long time, using with deep learning, and conventional method, for instance:

Syed, A.S.; Sierra-Sosa, D.; Kumar, A.; Elmaghraby, A. A Deep Convolutional Neural Network-XGB for Direction and Severity Aware Fall Detection and Activity Recognition. Sensors 202222, 2547. https://doi.org/10.3390/s22072547

  1. Iguchi, J. H. Lee and S. Okamoto, "Enhancement of Fall Detection Algorithm Using Convolutional Autoencoder and Personalized Threshold," 2021 IEEE International Conference on Consumer Electronics (ICCE), Las Vegas, NV, USA, 2021, pp. 1-5, doi: 10.1109/ICCE50685.2021.9427732.
  2. Cruciani et al., "Comparing CNN and Human Crafted Features for Human Activity Recognition," 2019 IEEE SmartWorld, Ubiquitous Intelligence & Computing, Advanced & Trusted Computing, Scalable Computing & Communications, Cloud & Big Data Computing, Internet of People and Smart City Innovation (SmartWorld/SCALCOM/UIC/ATC/CBDCom/IOP/SCI), Leicester, UK, 2019, pp. 960-967, doi: 10.1109/SmartWorld-UIC-ATC-SCALCOM-IOP-SCI.2019.00190.

And the similar method was also used in processing temporal data not limited to IMU signals, for example:

  1. Ye, Y. Li, R. Jin and Q. Li, "Towards Accurate Odor Identification and Effective Feature Learning With an AI-Empowered Electronic Nose," in IEEE Internet of Things Journal, doi: 10.1109/JIOT.2023.3299555.

Thus I would recommend the authors to give a more detailed review on the related works

<Response 2>

Thank you for mentioning exemplary papers. We apologize for any confusion caused by our use of the term "fall detection" without specifying "pre-impact." Our research specifically targets the pre-impact fall detection, which is more challenging than the post-fall detection due to the need to detect falls before impact. We believe there is a significant difference between the pre-impact and the post-fall detection. While post-fall detection can achieve high accuracy with simple DNNs and does not necessitate a specific segmentation technique, pre-impact fall detection requires a precise definition of the start and end of a fall, and also demands a superior model for effective fall detection. This necessity is emphasized by Musci [15] and Yu [18], who provide fall annotations in their datasets. We acknowledge that deep learning research in motion recognition using inertial sensors has been ongoing for a while, but the application contexts would be different. The papers you mentioned were post-fall detection, human activity recognition, and electronic nose research, which are slightly different from our research focus. However, we have added a commented review on deep learning-based post-fall detection in order to enhance reader understanding.

<Revised manuscript 2>

Some studies tried to detect falls using deep learning. Syed et al.,[5] proposed a Convolutional Neural Network (CNN)-XGB Network for fall detection and activity recognition, considering the direction and severity of falls, achieving an unweighted average recall of 88%. Iguchi et al.,[6] employed a Convolutional Autoencoder for unsupervised fall detection, demonstrating an accuracy of 98%. Zhang et al.,[7] used CNN to recognize 17 movements, including falls, achieving an identification rate of 91.5%. These studies focus on post-fall detection, including impact values during falls. The binary classification accuracy is nearly 100%, with ongoing research in the direction of multi-class classification. However, post-fall detection algorithms can quickly detect falls through alarms, but they cannot prevent the impact of falls itself. (Line 37~46)
